## [Transparent Peer Review file · Nature Communications]

Maximizing energy utilization and lithium leaching efficiency via sequential electrochemical dual-oxidation and soaking-relaxation

Corresponding Author: Professor Hong Chen

Version 0:

Reviewer comments:

Reviewer #1

(Remarks to the Author)

This work proposes an energy-efficient two-stage continuous oxidation method that improves current efficiency and lithium selectivity. The mechanism of lithium ion leaching during this process was elucidated in detail. However, further refinements and clarifications are needed to meet publication standards. Detailed comments and suggestions are provided below:

1. Can the authors clarify how the results indicating that OER caused an additional 19.97% and 26.86% increase in electric energy input were obtained (Line 131)? Is the observed decrease in current efficiency partially attributable to the oxidation of lattice oxygen?
2. Figure 4e demonstrates that lithium extraction efficiency in KCl solution is significantly higher than that observed in NaCl solution. Why, then, was potassium not selected for subsequent investigations of ionic concentration conditions?
3. The finding that “the L3/L2 ratio of Co initially decreased from 2.45 to 2.21 at Stage I and subsequently increased to 2.33 at Stage II, suggesting reduction followed by oxidation of Co” appears inconsistent with established principles. Does a decrease in the Co L3/L2 ratio not typically indicate oxidation rather than reduction?
4. The authors assert that “the reduction of Co results from high-valence Co oxidizing lattice oxygen under high-voltage conditions during Stage I.” However, in Figure 5e, the Co²⁺ content increases while the Co³⁺ content decreases in Stage I. Does this imply that some Co³⁺ oxidized lattice oxygen, contradicting the subsequent oxidation of Co²⁺ in Stage II?
5. How can On⁻-driven ion exchange in Stage II be experimentally verified, and what is its underlying mechanistic basis?
6. Does the oxidation of Ni and Co by On⁻ in Stage II facilitate lithium leaching?

Reviewer #2

(Remarks to the Author)

The manuscript presents a two-stage electrochemical strategy combining electrochemical dual-oxidation (EDO) and soaking-relaxation to enhance lithium leaching efficiency from spent NCM-type lithium-ion battery (LIB) cathodes. While the study is experimentally thorough and includes a range of characterization techniques (XRD, XPS, EELS, Raman, SEM), it does not meet the novelty and conceptual significance required for publication in Nature Communications. The core concepts—dual oxidation, redox-active electrolytes, and post-treatment soaking—are all well-established in the field. The work offers process optimization but not a new mechanistic framework, electrochemical principle, or materials platform. The authors claim conceptual innovation by introducing a two-stage continuous oxidation process for lithium leaching. However, similar methodologies have been reported across multiple studies: Gu et al. (2024) demonstrated a dual-oxidation electrochemical process for lithium recovery from seawater using Cl⁻ radicals and electrochemical tuning (PNAS, 121, e2414741121); Dang et al. (2022) employed Cl⁻-mediated radical systems in electro-oxidation for Li⁺ recovery (Chem. Eng. J., 435, 135169); Lv et al. (2021) used an electric field with relaxation leaching to enhance extraction yield (Appl. Catal. B: Environ., 283, 119634). The addition of a soaking-relaxation step as a standalone “stage II” is neither novel nor sufficiently mechanistically distinct. Moreover, the authors overstate their originality and fail to situate the method within the existing body of literature. While the manuscript reports extensive characterization, the interpretations are speculative in several places. The presence of On⁻ species is inferred but never directly quantified, and the conclusion that these species drive ion exchange in Stage II lacks direct evidence. The use of KI to suppress leaching is not a definitive indicator of On⁻ involvement, as other oxidants or residual Cl₂ could also react. The “quantitative breakdown” of driving mechanisms (Fig. 6b) is based on a closed calculation in Supplementary Note 5 and lacks reproducibility. There is a conflation of chemical

intuition with mechanistic proof, which undermines the scientific rigor of the claims. The techno-economic model is overly optimistic and omits critical factors: gas evolution (Cl₂) is hazardous and not addressed in scaling projections; waste treatment, electrolyte reuse, and metal recovery are not accounted for; claims of gross profit (\$379.68/ton) and electricity savings (45.29%) are based on lab-scale assumptions with unclear generalizability. The study lacks a clear scale-up pathway, casting doubt on the practicality of the method for industrial lithium recovery. The manuscript is overly long and often redundant in its explanations (especially in Section 2). The use of terms such as "unit lithium leaching efficiency" is awkward and should be revised for clarity. Figures (e.g., Fig. 1c, Fig. 6a) exaggerate redox roles during soaking, which contradicts the text and analytical data (EPR, DPD). The literature review in the introduction is incomplete and omits several critical prior works. Reference formatting is inconsistent in the bibliography. More compact data summarization in figures/tables and streamlining of redundant experimental details are also recommended. Among the most relevant prior works undermining the novelty of the manuscript are: Yang et al. (2023) – Environ. Sci. Technol. (<https://doi.org/10.1021/acs.est.2c08735>), which demonstrated direct electrochemical leaching with >95% efficiency from NCM cathodes; Dang et al. (2022) – Chem. Eng. J. (<https://doi.org/10.1016/j.cej.2022.135169>), which employed a dual oxidation strategy; Gu et al. (2024) – PNAS (<https://doi.org/10.1073/pnas.2414741121>), which combined dual electrochemical oxidation with soaking; Yu et al. (2019) – Energy Environ. Sci. (<https://doi.org/10.1039/C9EE01255H>), which proposed redox-targeted lithium recycling; Lv et al. (2021) – Appl. Catal. B (<https://doi.org/10.1016/j.apcatb.2020.119634>), which explored electric field-assisted lithium recovery; and Liu et al. (2020) – ACS Appl. Energy Mater. (<https://doi.org/10.1021/acsaem.0c00633>), which examined energy-tuned electrochemical extraction including a soaking phase. While the manuscript includes detailed experimentation and characterization, it does not present a novel concept, transformative mechanism, or breakthrough process. The dual-oxidation and soaking-relaxation framework is already well-established in the literature, and the manuscript's incremental contribution does not meet the threshold for Nature Communications.

Reviewer #3

(Remarks to the Author)

This manuscript presents a two-stage continuous lithium oxidative leaching method with optimized energy input for recovering lithium from spent NCM cathode materials. Comprehensive characterizations have been conducted to elucidate the interfacial mechanisms involved throughout the two-stage leaching process. An interesting discovery regarding the role of oxidized lattice oxygen in driving ion exchange to achieve in-depth lithium leaching has been unraveled. This approach achieved an optimal lithium leaching efficiency of over 98% while reducing energy consumption by 45.29%. This work represents a significant advancement in sustainable battery recycling, successfully addressing critical challenges of energy efficiency and selectivity through an electrochemical oxidation process. Congratulations to the authors for setting a novel benchmark in the field of electrochemically driven sustainable lithium recycling. Thus, here I suggest publishing it in Nature Communications after minor revisions.

Below are a few suggestions to further enhance the manuscript:

1. In the introduction, consider adding a sentence comparing the proposed method with a single electrochemical dual-oxidation (EDO) approach to better highlight the advantages of the two-stage method.
2. Please unify the terms "electrochemical dual-oxidation (EDO)" and "dual-oxidation" throughout the manuscript to avoid confusion. For instance, in line 67, line 100, and Fig. 6z, these terms should be consistently revised to either "electrochemical dual-oxidation" or "EDO."
3. The sentence in line 78 needs further clarification. It would be better phrased as: "Through the synergy of lithium leaching in stages I and II, both the electric energy conversion efficiency and the lithium leaching efficiency have been maximized."
4. In the EELS spectra presented in Figs. 5a–c, the labeling of stages should be consistent with the XPS figures. For instance, "I" and "I + II" should be revised to "stage I" and "stage II," respectively.
5. Given the mention of lattice oxygen participation during the oxidation process, why were the transition metals such as Co and Ni not oxidized to higher valence states, which could then contribute to the relaxed-stage leaching?
6. In Supplementary Table 9, the units (\$) should be included in the last three rows (Total cost, Product revenue, and Total gross profit).
7. Could the authors elaborate on how the two-stage leaching process would be integrated into a continuous mode for future industrial-scale applications?

Version 1:

Reviewer comments:

Reviewer #1

(Remarks to the Author)

The authors have solved the proposed issues; thus, I recommend the acceptance of this manuscript.

Reviewer #3

(Remarks to the Author)

For my part, all my concerns have been addressed.

I have also reviewed the questions raised by reviewer #2 as well as the authors' response. I feel that these concerns have also been satisfactorily addressed.

Therefore, I suggest the publication of this manuscript in its current form.

Point-by-point Response Letter

Dear Respected Editor and Reviewers,

We sincerely thank the editor for handling our manuscript and providing us with this valuable opportunity to revise it. We are also extremely grateful to all the reviewers for their hard work and for providing constructive comments on our manuscript, **"Maximizing energy utilization and lithium leaching efficiency via sequential electrochemical dual-oxidation and soaking-relaxation" [NCOMMS-25-43469]**.

We have carefully considered all these comments, which have been greatly helpful in improving our manuscript. We took all suggestions seriously and carefully revised the original manuscript to enhance and polish both its scientific content and writing.

Herein, we provide a point-by-point response letter together with the revised version of the manuscript in tracking mode. All changes have been highlighted in red.

We sincerely hope that the revised manuscript meets the high standards of all the reviewers and editors for re-review.

Thank you very much for your time and consideration.

Best wishes.

Here below is our point-by-point reply to all the reviewers' comments

Reviewer: 1

General comment: This work proposes an energy-efficient two-stage continuous oxidation method that improves current efficiency and lithium selectivity. The mechanism of lithium ion leaching during this process was elucidated in detail. However, further refinements and clarifications are needed to meet publication standards. Detailed comments and suggestions are provided below:

Response: We sincerely appreciate the reviewer's insightful comments and constructive suggestions, which have significantly improved the clarity and rigor of our manuscript. Below, we provide detailed responses to each comment:

1. Can the authors clarify how the results indicating that OER caused an additional 19.97% and 26.86% increase in electric energy input were obtained (Line 131)? Is the observed decrease in current efficiency partially attributable to the oxidation of lattice oxygen?

Response: Thank you very much for your insightful question. The additional electric energy input (19.97% and 26.86%) was calculated by comparing the actual energy consumption determined based on the measured charge quantity (Q) and applied voltage (U) with the theoretical energy required for lithium leaching (see Supplementary Notes 1 and 2 for details). Specifically, the electric energy input for 55 min-EDO and 60 min-EDO corresponded to 119.97% and 126.86% of the theoretical energy input, respectively. Therefore, the portions exceeding 100% (i.e., 19.97% in 119.97% and 26.86% in 126.86%), represent the additional energy input.

We fully agree that this excess energy input primarily originates from competing side reactions occurring at the electrode-electrolyte interface, including the chlorine evolution reaction (CER), the oxidation of lattice oxygen reaction, and oxygen evolution reaction (OER). All these side reactions consume the charge intended for lithium leaching. However, side reactions, such as lattice oxygen oxidation and CER, indirectly assist lithium leaching during the subsequent relaxation stage. Therefore, a core focus of this manuscript is to investigate these reactions and optimize the energy input for maximum efficiency. Given the importance of this challenge in electrochemical lithium leaching, we consider this work to be a pioneering investigation into this area.

Thank you for raising this point; it highlighted that our initial manuscript lacked a comprehensive list of these side reactions. We have now provided a corresponding detailed discussion within the manuscript.

The revised text is shown in line 122 as follows:

"The unavoidable occurrence of side reactions, including lattice oxygen oxidation, chloride evolution reaction and oxygen evolution reaction, resulting in an additional consumption of the input electric energy."

2. Figure 4e demonstrates that lithium extraction efficiency in KCl solution is significantly higher than that observed in NaCl solution. Why, then, was potassium not selected for subsequent investigations of ionic concentration conditions?

Response: Thank you for your insightful question. As shown in Fig. 4e, lithium leaching efficiency was initially higher with the KCl solution, but this difference diminished in the later stage. Despite the initial advantage of KCl, we opted to use NaCl for subsequent investigations due to considerations of material cost for future industrial applications. Specifically, the price of NaCl is substantially lower than that of KCl (for instance, in Sigma, the ACS reagent NaCl costs \$ 91/kg, while KCl costs \$162/kg). Our ultimate goal is to use naturally abundant seawater or waste saline water as the electrolyte in industrial-scale applications. We acknowledge that with technological advancements and potential cost reductions, KCl may also become a viable electrolyte option in the future.

3. The finding that "the L₃/L₂ ratio of Co initially decreased from 2.45 to 2.21 at Stage I and subsequently increased to 2.33 at Stage II, suggesting reduction followed by oxidation of Co" appears inconsistent with established principles. Does a decrease in the Co L₃/L₂ ratio not typically indicate oxidation rather than reduction?

Response: Thank you for your insightful question. You are correct that, under typical circumstances, a decrease in the L₃/L₂ ratio often indicates oxidation. However, changes in oxidation state are more directly correlated with the shifts in the L₃ and L₂ peak positions themselves. In our original manuscript, our analysis of Co EELS spectra solely based on the L₃/L₂ ratio was incorrect. Furthermore, our analysis of the Ni and Mn EELS spectra was incomplete. We apologize for these oversights and sincerely appreciate you bringing this critical issue to our attention. We have now re-analyzed and revised the relevant content in the manuscript accordingly.

The revised text is shown in line 287 as follows:

"The EELS analysis results are summarized in Fig. 5a-c and Supplementary Fig. 10. Throughout the entire two-stage continuous oxidation lithium leaching process, the L_3 and L_2 peaks of Ni blue shifted, with peak intensity ratio decreased from 3.33 to 2.46, confirming continuous Ni oxidation (Fig. 5a). Co, however, exhibited a more complex behavior (Fig. 5b). The L_3 and L_2 peaks showed significant blue shifts in Stage I but then red shifts in Stage II. When viewed alongside the Co XPS data (Fig. 5e), this suggests that Co undergoes an initial reduction followed by subsequent oxidation. The initial reduction is linked to the oxidation of lattice oxygen by high-valence Co under the high voltage of Stage I.^{31,32} Notably, the L_3/L_2 ratio did not follow the same simple trend (decreasing from 2.45 to 2.21, then increasing to 2.33). This complex pattern is likely due to the simultaneous occurrence of sequence redox reaction and local structure evolution for Co, as has been reported in battery research community by Jiang³³ and Guo³⁴. Finally, Mn remained highly stable. Its L_3 and L_2 peak positions and L_3/L_2 ratio barely changed throughout the process (Fig. 5c). This stability can be attributed to the role of Mn in maintaining the structural integrity of the material, with minimal contribution to the charge storage.³⁵"

31 Shen, Y. et al. Sodium doping derived electromagnetic center of lithium layered oxide cathode materials with enhanced lithium storage. *Nano Energy* **94**, 106900 (2022).

32 House, R. A. et al. The role of O_2 in O-redox cathodes for Li-ion batteries. *Nature Energy* **6**, 781-789 (2021).

33 Jiang, Q. et al. Surface oxygenate species on TiC reinforce cobalt-catalyzed fischer-tropsch synthesis. *ACS Catal.* **11**, 8087-8096 (2021).

34 Guo, Y. et al. In situ exsolved CoFe alloy nanoparticles for stable anodic methane reforming in solid oxide electrolysis cells. *Joule* **8**, 2016-2032 (2024).

To more clearly illustrate the shifts in the L_3 and L_2 peak positions, we have revised the EELS spectra accordingly (Fig. 5a-c). The updated figure is presented below:

Fig. 5 Chemical and electronic state evolution of Ni, Co, and Mn during two-stage continuous oxidation lithium leaching process. **Ex-situ electron energy loss spectra of (a) Ni, (b) Co, and (c) Mn; Ex-situ X-ray photoelectron spectra of (d) Ni 2p, (e) Co 2p, and (f) Mn 2p; (g) XPS spectra of O1s; (h) O K-edge of EELS spectra; (i) Raman spectra.**

In summary, despite the complex nature of the redox process, it is evident that the electrochemical dual-oxidation (Stage I) involves the oxidation of lattice oxygen (O^{2-} to O^{n-} , where $n < 2$), alongside the concurrent oxidation of both Ni and Co. However, Co exhibits unique behavior: the electronic energy levels of Co^{3+} overlap with those of lattice oxygen, and under applied voltage, Co^{3+} facilitates the oxidation of lattice oxygen, resulting in the generation of Co^{2+} . This phenomenon has been frequently reported in studies of lithium-rich cathode materials, as seen in works by House et al. (2021) in Nature Energy (<https://doi.org/10.1038/s41560-021-00780-2>) and Qin et al. (2022) in Advanced Energy Materials (<https://doi.org/10.1002/aenm.202201549>). Specifically, in stage I, the oxidation of Co^{2+} and the reduction of Co^{3+} are competing reactions; because the reduction of Co^{3+} is more substantial than the oxidation of Co^{2+} ,

the overall valence state of Co decreases, as observed in both the Co EELS and XPS spectra at stage I.

4. The authors assert that "the reduction of Co results from high-valence Co oxidizing lattice oxygen under high-voltage conditions during Stage I." However, in Figure 5e, the Co^{2+} content increases while the Co^{3+} content decreases in Stage I. Does this imply that some Co^{3+} oxidized lattice oxygen, contradicting the subsequent oxidation of Co^{2+} in Stage II?

Response: Thank you for the insightful question. As shown in Fig. 5e, the Co^{2+} content increases while the Co^{3+} content decreases during Stage I. This indeed indicates that a portion of Co^{3+} oxidizes lattice oxygen. Due to the overlap of the electronic energy levels between Co^{3+} and lattice oxygen, the applied voltage facilitates the oxidation of lattice oxygen, resulting in the generation of Co^{2+} . This phenomenon has been frequently reported in studies of lithium-rich cathode materials, such as House et al. (2021)-Nature Energy (<https://doi.org/10.1038/s41560-021-00780-2>) and Qin et al. (2022)-Advanced Energy Materials (<https://doi.org/10.1002/aenm.202201549>). Furthermore, the oxidation of lattice oxygen by Co^{3+} occurs during Stage I under an applied voltage, whereas the oxidation of Co^{2+} takes place in Stage II in the absence of voltage. Thus, these two phenomena are not contradictory.

5. How can O^{n-} -driven ion exchange in Stage II be experimentally verified, and what is its underlying mechanistic basis?

Response: Thank you for your insightful question. The overall lithium leaching process within this project is an ion-exchange process. The underlying mechanistic basis is the framework electron density resulting from ion-exchangeable interlayer cation exchange. To experimentally verify this process, we carefully illustrated the data with the corresponding discussion in Section 2.4. Quantitative analysis using ICP-MS and STEM-EDS confirmed that an equimolar ion exchange between Na^+ and Li^+ occurs during stage II. We then further investigated the driving force behind this ion-exchange reaction. Specifically, XPS spectra of O 1s and the newly added O K-edge EELS spectra (Fig. 5h) demonstrate the generation of O^{n-} ($n < 2$) during stage I and its subsequent consumption in stage II. Meanwhile, the EELS and XPS analysis results of Ni and Co indicate that their valence states increase in stage II. Since there is no external driving force for redox reactions in stage II, and the only species with reduced valence states

within the material is O^{n-} , this proves that O^{n-} acts as the oxidizing agent. Furthermore, we conducted a supplementary experiment by introducing KI, a strong reducing agent capable of rapidly eliminating oxidants. The results further confirm that the ion-exchange reaction between Na^+ and Li^+ requires an oxidizing agent driving force. Besides, we provide extra time-dependent O K-edge EELS spectra within Fig. 5h.

Fig. 5h O K-edge of EELS spectra.

6. Does the oxidation of Ni and Co by O^{n-} in Stage II facilitate lithium leaching?

Response: Thank you for your question. Yes, the oxidation of Ni and Co by O^{n-} indeed occurs during the O^{n-} -driven ion-exchange process, and this redox reaction synergistically contributes to lithium leaching as we explained in previous questions.

We are grateful for all the excellent comments and suggestions provided above. These recommendations have significantly improved the logic and clarity of the manuscript, enhancing the overall persuasiveness of our research. Your expert input and valuable time have proven crucial to refining this paper. Thank you once again for your meticulous review and guidance.

Reviewer: 2

General comment: The manuscript presents a two-stage electrochemical strategy combining electrochemical dual-oxidation (EDO) and soaking-relaxation to enhance lithium leaching efficiency from spent NCM-type lithium-ion battery (LIB) cathodes. While the study is experimentally thorough and includes a range of characterization techniques (XRD, XPS, EELS, Raman, SEM), it does not meet the novelty and conceptual significance required for publication in Nature Communications. The core concepts—dual oxidation, redox-active electrolytes, and post-treatment soaking—are all well-established in the field. The work offers process optimization but not a new mechanistic framework, electrochemical principle, or materials platform. The authors claim conceptual innovation by introducing a two-stage continuous oxidation process for lithium leaching. However, similar methodologies have been reported across multiple studies: Gu et al. (2024) demonstrated a dual-oxidation electrochemical process for lithium recovery from seawater using Cl^\cdot radicals and electrochemical tuning (PNAS, 121, e2414741121); Dang et al. (2022) employed Cl^\cdot -mediated radical systems in electro-oxidation for Li^+ recovery (Chem. Eng. J., 435, 135169); Lv et al. (2021) used an electric field with relaxation leaching to enhance extraction yield (Appl. Catal. B: Environ., 283, 119634). The addition of a soaking-relaxation step as a standalone "stage II" is neither novel nor sufficiently mechanistically distinct. Moreover, the authors overstate their originality and fail to situate the method within the existing body of literature. While the manuscript reports extensive characterization, the interpretations are speculative in several places. The presence of $\text{O}^{\cdot-}$ species is inferred but never directly quantified, and the conclusion that these species drive ion exchange in Stage II lacks direct evidence. The use of KI to suppress leaching is not a definitive indicator of $\text{O}^{\cdot-}$ involvement, as other oxidants or residual Cl_2 could also react. The "quantitative breakdown" of driving mechanisms (Fig. 6b) is based on a closed calculation in Supplementary Note 5 and lacks reproducibility. There is a conflation of chemical intuition with mechanistic proof, which undermines the scientific rigor of the claims. The techno-economic model is overly optimistic and omits critical factors: gas evolution (Cl_2) is hazardous and not addressed in scaling projections; waste treatment,

electrolyte reuse, and metal recovery are not accounted for; claims of gross profit (\$379.68/ton) and electricity savings (45.29%) are based on lab-scale assumptions with unclear generalizability. The study lacks a clear scale-up pathway, casting doubt on the practicality of the method for industrial lithium recovery. The manuscript is overly long and often redundant in its explanations (especially in Section 2). The use of terms such as "unit lithium leaching efficiency" is awkward and should be revised for clarity. Figures (e.g., Fig. 1c, Fig. 6a) exaggerate redox roles during soaking, which contradicts the text and analytical data (EPR, DPD). The literature review in the Introduction is incomplete and omits several critical prior works. Reference formatting is inconsistent in the bibliography. More compact data summarization in figures/tables and streamlining of redundant experimental details are also recommended. Among the most relevant prior works undermining the novelty of the manuscript are: Yang et al. (2023) – *Environ. Sci. Technol.* (<https://doi.org/10.1021/acs.est.2c08735>), which demonstrated direct electrochemical leaching with >95% efficiency from NCM cathodes; Dang et al. (2022) – *Chem. Eng. J.* (<https://doi.org/10.1016/j.cej.2022.135169>), which employed a dual oxidation strategy; Gu et al. (2024) – *PNAS* (<https://doi.org/10.1073/pnas.2414741121>), which combined dual electrochemical oxidation with soaking; Yu et al. (2019) – *Energy Environ. Sci.* (<https://doi.org/10.1039/C9EE01255H>), which proposed redox-targeted lithium recycling; Lv et al. (2021) – *Appl. Catal. B* (<https://doi.org/10.1016/j.apcatb.2020.119634>), which explored electric field–assisted lithium recovery; and Liu et al. (2020) – *ACS Appl. Energy Mater.* (<https://doi.org/10.1021/acsaem.0c00633>), which examined energy-tuned electrochemical extraction including a soaking phase. While the manuscript includes detailed experimentation and characterization, it does not present a novel concept, transformative mechanism, or breakthrough process. The dual-oxidation and soaking–relaxation framework is already well-established in the literature, and the manuscript's incremental contribution does not meet the threshold for *Nature Communications*.

Response: We sincerely appreciate your comprehensive summary of our research and the constructive suggestions provided, which have greatly guided us in further

improving the manuscript. Once again, we are grateful for your recognition of the experimental integrity of our research. We have carefully considered and incorporated your valuable comments into the revisions, providing detailed explanations for the points of misunderstanding. We hope that the revised version meets the high standards of Nature Communications.

1. The core concepts—dual oxidation, redox-active electrolytes, and post-treatment soaking—are all well-established in the field. The work offers process optimization but not a new mechanistic framework, electrochemical principle, or materials platform. However, similar methodologies have been reported across multiple studies: Gu et al. (2024) demonstrated a dual-oxidation electrochemical process for lithium recovery from seawater using Cl^\cdot radicals and electrochemical tuning (PNAS, 121, e2414741121); Dang et al. (2022) employed Cl^- -mediated radical systems in electro-oxidation for Li^+ recovery (Chem. Eng. J., 435, 135169); Lv et al. (2021) used an electric field with relaxation leaching to enhance extraction yield (Appl. Catal. B: Environ., 283, 119634). The addition of a soaking-relaxation step as a standalone "stage II" is neither novel nor sufficiently mechanistically distinct. Moreover, the authors overstate their originality and fail to situate the method within the existing body of literature.

Response: Thank you for your comment. The concept and mechanism of dual-oxidation that combines electrode oxidation with electrocatalytic electrolyte oxidation was proposed by Dr. Gu in our former research when we employed LiFePO_4 as an example to recycle lithium (Gu et al. (2024)-PNAS, (<https://doi.org/10.1073/pnas.2414741121>)). In contrast, the post-treatment soaking mechanism is detailed and illustrated for the first time within this work. Actually, we have properly cited our previous work in line 70 and highlighted it in our initial submitted manuscript (citation number 17).

One important discovery is that, based on our previous study, we noticed that although a substantial amount of electric energy is consumed, solely employing the dual-oxidation method, the lithium leaching efficiency is difficult to exceed 95%. This somehow inspires us that the dual-oxidation method presents significant challenges in balancing energy input and lithium leaching efficiency. Therefore, in this work, we devote considerable effort to studying the relationship between energy input and lithium leaching efficiency. Eventually, here for the first time, we propose a technique that combines electrochemical dual-oxidation (stage I) with a soaking-relaxation process

(stage II) to fine-tune the relationship between energy input and lithium leaching with $\text{LiNi}_{1/3}\text{Co}_{1/3}\text{Mn}_{1/3}\text{O}_2$ as a typical example. Following the excellent performance of this integrated method, we conducted an in-depth investigation into the underlying mechanisms. The proposed two-stage method, which optimizes energy input and lithium leaching efficiency, together with the underlying in-depth reaction mechanism, are three crucial academic contributions within this work that significantly differ from our previous report and other literature.

From a mechanistic point of view, specifically, in stage I (electrochemical dual-oxidation), in addition to the previously reported electrode oxidation and electrocatalytic electrolyte oxidation mechanisms proposed in our PNAS paper. In this study, we reveal a newly identified electrochemical-assisted Li^+/Na^+ exchange mechanism in $\text{LiNi}_{1/3}\text{Co}_{1/3}\text{Mn}_{1/3}\text{O}_2$, which contributes up to 15.97% of the overall lithium leaching. To the best of our knowledge, this has not been reported in previous electrochemical lithium leaching studies. This discovery represents a mechanistic innovation within the framework of electrochemical dual oxidation, which is subsequently applicable to our newly proposed stage II reaction (soaking process). We appreciate the references you provided. However, upon careful study, we found that no existing studies have reported the leaching behavior during soaking in the references you provided or other databases. Even the critical role of active species has been rarely studied, except in our paper published in PNAS. For instance, although Dang's work (Chem. Eng. J., 435, 135169) employed a Cl^- -containing electrolyte, the role of Cl^- in the lithium leaching mechanism was not investigated and explained; it merely served as an ionic conductor. Similarly, Lv's work (Appl. Catal. B: Environ., 283, 119634) also utilized a Cl^- -containing electrolyte solely as an ionic conductor, and no soaking relaxation process was explained in their study. In our previous PNAS paper, to study the Li leaching behavior from LiFePO_4 , we hastily hypothesized that oxidizing chlorine species generated during electrochemical dual-oxidation could enhance lithium leaching, without conducting a detailed mechanistic study. So this paper takes another complex composition $\text{LiNi}_{1/3}\text{Co}_{1/3}\text{Mn}_{1/3}\text{O}_2$ cathode as an example, not only fills the critical energy-lithium leaching efficiency optimization gap, but also unraveling the more fruitful reaction pathways that drive in-depth lithium leaching with minimal energy input, which is crucial for future application of electrochemical method for critical metal leaching from spent LIBs cathodes and beyond solid wastes or original minerals. Furthermore, in section 2.4, we conducted a detailed investigation of phase

transitions occurring during the two-stage continuous oxidation lithium leaching process using techniques such as in situ XRD and SEM. The XRD data were further refined to clarify the complex phase evolution (O_3 - P_2 - O_2) associated with lithium extraction.

Furthermore, we wish to emphasize that this work pioneers the proposal of a two-stage leaching method and the subsequent fine-tuning of energy input for lithium extraction. Our combined two-stage strategy significantly reduces the required energy input by 49.78% compared to previously reported electrochemical methods. We believe that achieving a 49.78% energy saving alongside a higher lithium leaching efficiency is a significant scientific advancement. Given that this is the first work to systematically optimize the critical parameters of energy input and lithium leaching, which are essential for future engineering applications of electrochemical methods, and that the in-depth reaction mechanism is well-illustrated. We are confident that this is a valuable concept with solid in-depth mechanism illustration work, which is not easily found in the resource recycling and pollution control research community, and should merit publication in Nature Communications.

2. While the manuscript reports extensive characterization, the interpretations are speculative in several places. The presence of O^{n-} species is inferred but never directly quantified, and the conclusion that these species drive ion exchange in Stage II lacks direct evidence. The use of KI to suppress leaching is not a definitive indicator of O^{n-} involvement, as other oxidants or residual Cl_2 could also react. The "quantitative breakdown" of driving mechanisms (Fig. 6b) is based on a closed calculation in Supplementary Note 5 and lacks reproducibility. There is a conflation of chemical intuition with mechanistic proof, which undermines the scientific rigor of the claims.

Response: Thank you for your insightful comments. The evidence for the presence of O^{n-} species, based solely on XPS characterization in our previous manuscript, is indeed insufficient. Therefore, we have supplemented our analysis with O K-edge EELS data, as shown in Fig. 5h. Regarding the analysis of the O K-edge EELS spectra, the revised text is shown in line 318 as follows:

"The O K-edge EELS (Fig. 5h) revealed that the pre-peak initially shifted to lower energy loss and then to higher energy loss during lithium leaching, accompanied by a corresponding increase and subsequent decrease in its relative intensity to the main peak, further supporting a sequence of lattice-oxygen (O^{2-}) oxidation in stage I followed

by reduction in stage II."

Indeed, current analytical techniques do not allow for the direct quantification of O^{n-} species. However, as shown in Fig. 6b and based on the quantitative evaluation of lithium leaching contributions detailed in Supplementary Note 5, we provide an indirect means of estimating the presence and impact of O^{n-} species. Furthermore, the initial purpose of conducting KI-inhibited lithium leaching experiments was to assist in validating the presence of O^{n-} species. We acknowledge that this approach serves only as supporting evidence and cannot be considered direct proof. But combining with XPS and EELS results, we believe that the evidence is strong enough to conclude the existence of O^{n-} . Furthermore, for your concern that KI inhibition alone cannot serve as a definitive indicator of residual O^{n-} species due to the potential influence of residual oxidizing species in the soaking solution. To address this concern, we replaced the post-electrochemical dual-oxidation soaking solution with a fresh 0.5 M NaCl solution prior to KI addition, thereby eliminating interference from residual oxidizing agents in the solution. The results showed no lithium leaching upon KI addition, confirming that KI's strong reducing capability eliminated the oxidizing species responsible for lithium extraction, further indicating the role of intrinsic oxidants within the material, such as O^{n-} species. As described in our response to reviewer 1's question 5, we provided detailed mechanistic evidence for the O^{n-} -driven ion exchange. First, we confirmed via ICP-MS and STEM-EDS that a stoichiometric exchange of Na^+ and Li^+ occurred during stage II. We then explored the driving force of this exchange. Through O 1s XPS spectra and newly supplemented O K-edge EELS data (Fig. 5h), we demonstrated the generation of O^{n-} ($n < 2$) during stage I and its subsequent consumption during stage II. Simultaneously, EELS and XPS analyses of Ni and Co revealed an increase in their oxidation states during stage II. Given that no external driving force exists during stage II and that O^{n-} is the only reducible species in the system, we conclude that O^{n-} serves as the sole oxidant facilitating ion exchange. The KI inhibition experiments, although auxiliary, further support that this ion exchange cannot proceed spontaneously and requires an oxidant.

The quantitative contributions to lithium leaching mechanisms presented in Figure 6b are indeed based on a closed-system calculation as detailed in Supplementary Note 5. It's important to note, however, that all estimations are derived from rigorous and scientifically validated experimental data. Our primary intention in performing this quantification was to provide a more comprehensive mechanistic understanding of the

process.

Finally, we thank you again for your insightful comments. They prompted us to consider this study with even more rigorous evidence.

Fig. 5h O K-edge of EELS spectra.

3. The techno-economic model is overly optimistic and omits critical factors: gas evolution (Cl_2) is hazardous and not addressed in scaling projections; waste treatment, electrolyte reuse, and metal recovery are not accounted for; claims of gross profit (\$379.68/ton) and electricity savings (45.29%) are based on lab-scale assumptions with unclear generalizability. The study lacks a clear scale-up pathway, casting doubt on the practicality of the method for industrial lithium recovery.

Response: Thank you for your valuable feedback on the techno-economic assessment section of our study. We have comprehensively revised the discussion regarding the scale-up possibility and techno-economic evaluation in the revised manuscript and supplementary information in accordance with your suggestions to more rigorously and thoroughly reflect the key factors in practical industrial applications. Below is our specific response to your concerns.

1. Regarding the safety and scaled-up handling of chlorine gas (Cl_2) emissions: We fully acknowledge the hazardous nature of Cl_2 and have explicitly addressed this in the revised manuscript by incorporating a dedicated gas treatment system. As detailed in Supplementary Note 8(6): 1) The Cl_2 generated during the EDO step will be neutralized via an alkaline scrubbing system using a sodium hydroxide (NaOH) solution, resulting in the formation of NaClO and NaCl. 2) The maximum amount of Cl_2 produced per ton of processed spent NCM111 material is estimated to be 31.87 kg, with an associated NaOH cost of \$4.22. 3) The capital investment and operational costs for this gas

treatment system have been included in our economic assessment (refer to Supplementary Table 9).

2. Regarding waste disposal, electrolyte reuse, and metal recovery. We have added the following to the economic assessment:

(1) Waste treatment: 1) Solid handling cost (Supplementary Note 8(7)): After the lithium leaching reaction is complete, the de-lithiated NCM111 must be scraped off the electrode plates. The de-lithiated NCM111 can be sold or further processed (this economic analysis does not consider any costs or revenues associated with the transition metals Ni, Co, and Mn). Assuming a labor cost of $\$0.05 \text{ kg}^{-1}$, the labor cost for handling one ton of spent NCM111 powder amounts to $\$50$. 2) Waste liquid treatment cost (Supplementary Note 8(8)): Waste liquid treatment cost: The leachate remaining after Li_2CO_3 recovery primarily contains Na^+ , Cl^- , and a slight excess of CO_3^{2-} and OH^- ions ($\text{pH} \approx 11.5$), with almost no heavy metal pollutant ions. Consequently, the treatment process is significantly simplified, requiring only a neutralization step to meet discharge standards. Using low-cost industrial hydrochloric acid (31%) for neutralization, it is estimated that treating 1 m^3 of this waste liquid requires 0.372 kg of acid. As processing one ton of spent NCM111 powder generates approximately 150 m^3 of leachate, the associated waste liquid treatment cost is $\$0.64$.

(2) Electrolyte reuse: To ensure process stability and high product quality, a conservative assumption of "single-use electrolyte" was adopted for the economic assessment in this study, which primarily focuses on validating the core efficiency of lithium recovery. Therefore, electrolyte recycling, which could be considered for future industrial scenarios, was not considered here.

(3) Scope of metal recovery: While this study focuses on lithium recovery, we explicitly state (at the beginning of Supplementary Note 8) that the delithiated NCM111 can be sold as a ternary precursor or further critical metal recycling. Its economic value has already been considered in the cost allocation.

3. Regarding the concerns about the generalizability of the gross profit and power-saving data based on lab-scale assumptions. Our revised analysis, detailed in Supplementary Notes 6-8, is now based on scale-up calculations using the experimental data from a 500-gram pilot-scale system. The economic assessment datasheet (Supplementary Table 9) has been updated accordingly: 1) Electricity savings: Revised from 45.29% to 49.78%, based on a comparison between the actual electrical charge input and the theoretical value. 2) Gross profit: Updated from $\$379.68 \text{ t}^{-1}$ to $\$644.16 \text{ t}^{-1}$.

1. This figure now comprehensively covers all costs, including materials, energy consumption, equipment depreciation, and waste treatment. (The primary reason for this increase is the substitution of potassium carbonate with sodium carbonate as the carbonate source for lithium carbonate recovery, reflecting the most common and economically viable choice in actual industrial practice.) 3) Equipment scale-up pathway: We have explicitly outlined that processing one ton of material per day would require 200 units of the customized equipment (designed for a 500-gram batch process). Detailed parameters regarding equipment investment, lifespan, and maintenance are provided in Supplementary Note 8(9).

4. For the scalability problem, we actually launched a startup company to commercialize our technique. The clarity of the industrialization pathway has been significantly enhanced in the revised manuscript. Specifically, Section 2.9 (Techno-economic evaluation) and Supplementary Note 8 now include: 1) Process flow diagrams (Fig. 7a-b) and photographs of the actual equipment (Supplementary Fig. 15a); 2) A scale-up model utilizing actual operational data; and 3) A precise definition of critical industrialization parameters, such as equipment scale, cycling time, energy consumption, and waste handling. These additions provide a clear and practical blueprint for industrial application.

Fig. 7 (a) Process flow diagram for Li_2CO_3 recovery; (b) Schematic diagram of the continuous-flow system used for Li_2CO_3 recovery.

Supplementary Fig. 15 (a) Photographs of the customized equipment and electrolytic cell.

We believe the revisions made have substantially improved the comprehensiveness and authenticity of the techno-economic assessment in reflecting the industrial viability of our method. We would like to express our sincere gratitude for your guidance, which has been instrumental in enhancing the rigor and practical utility of this paper. Thank you once again for your valuable comments.

4. The manuscript is overly long and often redundant in its explanations (especially in Section 2). The use of terms such as "unit lithium leaching efficiency" is awkward and should be revised for clarity. Figures (e.g., Fig. 1c, Fig. 6a) exaggerate redox roles during soaking, which contradicts the text and analytical data (EPR, DPD). The literature review in the Introduction is incomplete and omits several critical prior works. Reference formatting is inconsistent in the bibliography. More compact data summarization in figures/tables and streamlining of redundant experimental details are also recommended.

Response: Thank you for your valuable suggestions. In response, we have removed non-essential explanations, simplified redundant content, and eliminated unnecessary prepositional phrases. These revisions have been marked in the revised manuscript using strikethrough and red highlighting.

1. Deletion of non-essential explanations is as follows:

- 1) Therefore, their influence on lithium leaching efficiency during the soaking relaxation process can be negligible. (Line 270 of initial manuscript)
- 2) The valence states of transition metals and lattice oxygen changes further suggest that direct electrode oxidation occurred at stage I. (Line 337 of initial manuscript)
- 3) Herein, O^{n-} provides a critical driving force for the lithium leaching oxidation reaction at stage II. (Line 342 of initial manuscript)

4) The Raman spectra results further confirm that the driving force for the lithium leaching oxidation reaction in stage II originates from $O^{\cdot-}$ generation during the EDO process in stage I. (Line 348 of initial manuscript)

2. Simplification of redundant content are as follows:

1) "Different side reactions, including catalytic chloride oxidation, make achieving 100% lithium leaching efficiency hard, even with over 100% electric energy input. (Line 121 of initial manuscript)" and "The unavoidable occurrence of side reactions, such as OER led to 19.97% and 26.86% extra input electric energy consumption. (Line 130 of initial manuscript)" are simplified to "The unavoidable occurrence of side reactions, including lattice oxygen oxidation, chloride oxidation reaction, and oxygen evolution reaction, results in an additional consumption of the input electric energy. (Line 123 of revised manuscript)".

2) "in addition to the EDO lithium leaching, electrochemical-assisted Li^+/Na^+ ion exchange²⁴ also present, where P2-phase $Na_{0.01}Li_{0.47}Ni_{1/3}Co_{1/3}Mn_{1/3}O_2$ transforms into O2-phase $Na_{0.16}Li_{0.17}Ni_{1/3}Co_{1/3}Mn_{1/3}O_2$. (Line 206 of initial manuscript) is simplified to "both EDO leaching and electrochemical-assisted Li^+/Na^+ exchange²⁴ occurred, yielding O2-phase $Na_{0.16}Li_{0.17}Ni_{1/3}Co_{1/3}Mn_{1/3}O_2$. (Line 206 of revised manuscript)".

3) "In-situ powder XRD data have been collected to further elucidate the structural evolution dynamics during stage II, as shown in Fig. 3c. As the soaking relaxation process proceeds, the diffraction peaks shift to higher two-theta angles. Fig. 3d presents a magnified view of (002) diffraction peak regarding the O2-phase $Na_{0.16}Li_{0.17}Ni_{1/3}Co_{1/3}Mn_{1/3}O_2$. During the first 30 min of the soaking relaxation process, the diffraction peak shifts rapidly towards higher angles, whereas after 30 min, the shift becomes relatively slower. These results indicate a reduction in the interlayer spacing of the electrode material during stage II. To visualize the evolution of interlayer spacing, high-resolution transmission electron microscopy (HRTEM) images were collected before and after soaking relaxation, as shown in Supplementary Fig. 7. The interlayer spacing of the material decreased from 5.05 Å to 4.97 Å at stage II, consistent with the in-situ powder XRD observation. (Line 209 of initial manuscript)" is simplified to "In stage II, in-situ powder XRD (Fig. 3c) showed diffraction peaks shifting to higher angles. Fig. 3d magnifies the (002) peak of O2- $Na_{0.16}Li_{0.17}Ni_{1/3}Co_{1/3}Mn_{1/3}O_2$, showing rapid shifts in the first 30 min, followed by a slowing. This indicated a reduction in interlayer spacing, consistent with the HRTEM images (Supplementary Fig. 7), where the spacing decreased from 5.05 Å to 4.97 Å. (Line 208 of revised manuscript)".

- 4) "images revealed significant changes in Na content before and after soaking relaxation, confirming that Na^+ continuous intercalation occurred during stage II (Supplementary Fig. 8), consistent with the ICP-MS measurement data (Supplementary Fig. 6b). (Line 222 of initial manuscript)" is simplified to "(Supplementary Fig. 8) confirmed increased Na content during soaking, consistent with ICP-MS observation (Supplementary Fig. 6b). (Line 214 of revised manuscript)".
- 5) "In contrast, structural evolution facilitates the efficient lithium leaching process, demonstrating the importance of alkali metal insertions driving phase transitions, thus optimizing lithium extraction. (Line 247 of initial manuscript)" is simplified to ", while the subsequent structural transitions in turn facilitate efficient lithium leaching. (Line 239 of revised manuscript)".
- 6) "As shown in Fig. 4a, $\cdot\text{OH}$ and $\text{ClO}\cdot$ radicals were detected during the EDO process in stage I. However, no $\cdot\text{OH}$ or $\text{ClO}\cdot$ radical signals were observed at stage II, (Line 267 of initial manuscript) is simplified to "As shown in Fig. 4a, $\cdot\text{OH}$ and $\text{ClO}\cdot$ radicals were detected in stage I, but not in stage II, (Line 256 of revised manuscript)".
- 7) "The free chlorine content was also measured, as shown in Fig. 4b-c. The free chlorine content increased during stage I, indicating its generation through electrocatalytic oxidation of the electrolyte. In contrast, at stage II, the concentration gradually decreased with the extension of the soaking time. Eventually, its concentration reached zero after 3 hours, confirming its consumption in stage II lithium leaching process. (Line 272 of initial manuscript)" is simplified to "The free chlorine content (Fig. 4b-c) increased during stage I, but decreased to zero after 3 hours of soaking, confirming its consumption. (Line 258 of revised manuscript)".
- 8) "the lithium leaching efficiency after 3 hours was 98.32%, which is lower than the 99.87% achieved without solution replacement. The 1.55% efficiency difference in lithium leaching efficiency can be attributed to the contribution of residual free chlorine, suggesting that free chlorine participated in the lithium leaching reaction during stage II. (Line 279 of initial manuscript)" is simplified to "the lithium leaching efficiency after 3 hours dropped to 98.32%, ~1.55% lower than without replacement (99.87%), indicating that free chlorine contributed modestly to stage II leaching. (Line 263 of revised manuscript)".
- 9) "likely because no alkali cations were available in the solution for ion exchange reactions. In contrast, lithium leaching rates in the KCl solution were significantly higher than in the NaCl solution. (Line 286 of initial manuscript)" is simplified to

"whereas the KCl solution yielded higher leaching rates than the NaCl solution. (Line 268 of revised manuscript)".

10) "Ex-situ X-ray photoelectron spectroscopy (XPS) double confirms the valence state variation of Ni, Co, and Mn during the two-stage continuous oxidation lithium leaching process (Fig. 5d-f). (Line 317 of initial manuscript)" is simplified to "Ex-situ XPS double confirms the valence state evolution (Fig. 5d-f). (Line 302 of revised manuscript)".

11) "In addition, scanning kelvin probe (SKP) testing was conducted to study the changes in the corrosion potential of the material. As shown in Fig. 5g, the more negative potential difference indicates a higher corrosion potential of the material in the SKP images. The results indicate a significantly more negative potential after soaking has been preserved, indicating a notable increase in corrosion potential at stage II. This phenomenon may be related to the increased oxidation states of Ni and Co in NCM111, as higher oxidation states of transition metals can enhance the material's corrosion resistance, thereby affecting its corrosion potential. These findings align with the results of EELS and XPS studies, further confirming that Ni and Co in the material experienced an increase in oxidation state at stage II. (Line 320 of initial manuscript)" is simplified to "Scanning Kelvin probe (SKP) mapping (Supplementary Fig. 11) reveals a significantly more negative surface potential after soaking, indicating a notable change in the surface chemical potential/corrosion behavior at stage II. These SKP observations are consistent with the oxidation state changes inferred from EELS and XPS. (Line 303 of revised manuscript)".

12) "In previous work, we investigated the feasibility of the two-stage continuous oxidation lithium leaching method using commercial NCM111 as a model material. To check its performance in practical spent lithium-ion batteries, further leaching experiments have been conducted with spent NCM111. As shown in Supplementary Fig. 12a, the XRD pattern of the spent NCM111 reveals that it retains a well-defined layered structure with minimal impurities. SEM analysis was further conducted to observe the morphology of spent NCM111. Supplementary Fig. 12b-c reveals the presence of pores on the surfaces of some small particles, which may be attributed to the reaction between the material and the electrolyte during the battery's charge-discharge process, leading to surface corrosion. Supplementary Fig. 13a-b compares the lithium leaching efficiencies and corresponding energy input ratios under different EDO durations. (Line 383 of initial manuscript)" is simplified to "To validate the

universality and practical applicability of this two-stage strategy beyond model materials, leaching experiments were performed on spent NCM111 cathodes from practical spent lithium-ion batteries (Supplementary Fig. 13a-c). (Line 358 of revised manuscript)".

3. Deletion of unnecessary prepositional phrases:

1) during the two-stage continuous oxidation lithium leaching process. (Line 266 of initial manuscript)

2) during the redox process of lithium leaching. (Line 270 of initial manuscript)

3) to the overall lithium leaching efficiency. (Line 379 of initial manuscript)

The above revisions have reduced the total word count of the original manuscript by 481 words.

Additionally, we have revised the term "unit lithium leaching efficiency" to ">99% lithium leaching efficiency" and "100% lithium leaching efficiency" in line 92 and line 163 of the revised manuscript, respectively. Regarding your recommendation on the literature review in the Introduction, we carefully consulted all the references you provided and supplemented previously omitted citations. All of the above suggestions have been addressed and incorporated into the revised manuscript.

We acknowledge your concern regarding the potential overestimation of redox reactions during the soaking process based on the presented EPR and DPD data. We appreciate the opportunity to clarify this issue and address the misunderstanding. The EPR and DPD analyses indeed indicate that the oxidative contributions from free radicals and residual chlorine species during soaking account for less than 2% of the total lithium leaching efficiency. This limited contribution may have led to the impression that the role of redox reactions during soaking is exaggerated. However, it is essential to note that a more significant redox process occurring during soaking is the $O^{\bullet-}$ -driven ion exchange mechanism, which contributes as much as 15.43% to the overall lithium leaching rate. We have provided a detailed explanation of the $O^{\bullet-}$ -driven ion exchange mechanism in our previous responses and have revised the manuscript accordingly to enhance clarity. We apologize for any confusion caused by the initial presentation of the EPR and DPD data, which were intended solely to investigate the role of residual oxidative species in the electrolyte of several potential mechanisms involved.

5. Among the most relevant prior works undermining the novelty of the manuscript are:

Yang et al. (2023) – Environ. Sci. Technol. (<https://doi.org/10.1021/acs.est.2c08735>), which demonstrated direct electrochemical leaching with >95% efficiency from NCM cathodes; Dang et al. (2022) – Chem. Eng. J. (<https://doi.org/10.1016/j.cej.2022.135169>), which employed a dual oxidation strategy; Gu et al. (2024) – PNAS (<https://doi.org/10.1073/pnas.2414741121>), which combined dual electrochemical oxidation with soaking; Yu et al. (2019) – Energy Environ. Sci. (<https://doi.org/10.1039/C9EE01255H>), which proposed redox-targeted lithium recycling; Lv et al. (2021) – Appl. Catal. B (<https://doi.org/10.1016/j.apcatb.2020.119634>), which explored electric field–assisted lithium recovery; and Liu et al. (2020) – ACS Appl. Energy Mater. (<https://doi.org/10.1021/acsaem.0c00633>), which examined energy-tuned electrochemical extraction including a soaking phase. While the manuscript includes detailed experimentation and characterization, it does not present a novel concept, transformative mechanism, or breakthrough process. The dual-oxidation and soaking-relaxation framework is already well-established in the literature, and the manuscript's incremental contribution does not meet the threshold for Nature Communications.

Response: Thank you for providing the relevant references. We have carefully reviewed and analyzed the prior studies that you identified as potentially diminishing the novelty of our manuscript. Among them, the works by Yang et al. (2023) - Environ. Sci. Technol. (<https://doi.org/10.1021/acs.est.2c08735>), Dang et al. (2022) - Chem. Eng. J. (<https://doi.org/10.1016/j.cej.2022.135169>), and Lv et al. (2021) - Appl. Catal. B (<https://doi.org/10.1016/j.apcatb.2020.119634>) all report electrochemical lithium extraction methods that essentially rely on anodic oxidation, a process equivalent in principle to the conventional charging mechanism of lithium-ion battery cathodes. These studies do not present fundamentally new mechanisms for lithium extraction. We have already cited these works in lines 62 and 66 of our initial manuscript (The citation number are 9, 10, and 11). In contrast, the work by Gu et al. (2024) - PNAS (<https://doi.org/10.1073/pnas.2414741121>) - which focuses on LiFePO₄, introduces a new lithium extraction mechanism by identifying an additional contribution from electrocatalytic electrolyte oxidation, in parallel with anodic electrode oxidation. This led to the proposal of the dual-oxidation concept, which is mechanistically distinct from the previously reported electrode oxidation. We cited this work in line 69 of the initial manuscript (citation number 17). The study by Liu et al. (2020) - ACS Appl. Energy Mater. (<https://doi.org/10.1021/acsaem.0c00633>) primarily addresses the application of

lithium-ion cathode materials for fast-charging electric vehicles and is therefore not relevant to electrochemical lithium extraction research. Regarding Yu et al. (2019) – Energy Environ. Sci. (<https://doi.org/10.1039/C9EE01255H>), we were unable to find this reference in the database and therefore could not assess its relevance. In summary, among all the references reviewed, the work by Gu et al. represents the most comprehensive study to date in the field of electrochemical lithium extraction. In comparison, the two-stage continuous oxidation method proposed here, which fine-tunes the relationship between electron energy input and lithium leaching efficiency, is significantly different from the dual-oxidation method reported in our PNAS paper and other electrochemical lithium battery research.

Overall, besides the concept difference, the in-depth lithium leaching mechanism unravelled here, based on $\text{LiNi}_{1/3}\text{Co}_{1/3}\text{Mn}_{1/3}\text{O}_2$, is also significantly different from the previously reported one. We can confidently conclude that this work is a notable contribution to the recycling community, not only demonstrating new research concepts but also providing in-depth scientific insights and innovative equipment design to the electrochemical-driven battery recycling community. It does provide a benchmark or novel research approach to the sustainable community, as reviewer #3 mentioned, and could be expanded to other technical material recycling research. We believe that this work will inspire the next generation of researchers in electrochemical-driven critical metal recycling or the metallurgical engineering community and be worthy of publication in Nature Communications.

Finally, thank you for all the suggestions provided above. These comments have helped us improve the logic and clarity of the manuscript, as well as enhance the overall persuasiveness of the research. Your expert input and valuable time have been crucial in further refining this paper. Once again, we sincerely appreciate your time and guidance!

Reviewer: 3

General comment: This manuscript presents a two-stage continuous lithium oxidative leaching method with optimized energy input for recovering lithium from spent NCM cathode materials. Comprehensive characterizations have been conducted to elucidate the interfacial mechanisms involved throughout the two-stage leaching process. An interesting discovery regarding the role of oxidized lattice oxygen in driving ion exchange to achieve in-depth lithium leaching has been unraveled. This approach achieved an optimal lithium leaching efficiency of over 98% while reducing energy consumption by 45.29%. This work represents a significant advancement in sustainable battery recycling, successfully addressing critical challenges of energy efficiency and selectivity through an electrochemical oxidation process. Congratulations to the authors for setting a novel benchmark in the field of electrochemically driven sustainable lithium recycling. Thus, here I suggest publishing it in Nature Communications after minor revisions.

Response: We appreciate your kind comments on our research and your constructive suggestions. We've carefully considered your suggestions and revised the manuscript accordingly.

1. In the Introduction, consider adding a sentence comparing the proposed method with a single electrochemical dual-oxidation (EDO) approach to better highlight the advantages of the two-stage method.

Response: Thank you for your helpful suggestion. We have supplemented and refined the final paragraph of the Introduction to better highlight the advantages of our proposed method over the single electrochemical dual-oxidation method. The revised text is shown in line 80 as follows:

"Unlike the single-stage EDO method, which consumes high electric energy, our method maximizes both electric energy conversion efficiency and lithium leaching efficiency through the synergistic effects of stages I and II."

2. Please unify the terms "electrochemical dual-oxidation (EDO)" and "dual-oxidation" throughout the manuscript to avoid confusion. For instance, in line 67, line 100, and Fig. 6z, these terms should be consistently revised to either "electrochemical dual-oxidation" or "EDO".

Response: Thank you for your helpful suggestions. We have revised all terms throughout the text to make the wording clearer and more rigorous.

3. The sentence in line 78 needs further clarification. It would be better phrased as: "Through the synergy of lithium leaching in stages I and II, both the electric energy conversion efficiency and the lithium leaching efficiency have been maximized."

Response: Thank you for your helpful suggestion. We have revised the text in accordance with your suggestions.

4. In the EELS spectra presented in Figs. 5a-c, the labeling of stages should be consistent with the XPS figures. For instance, "I" and "I + II" should be revised to "stage I" and "stage II", respectively.

Response: Thank you for your helpful suggestion. We have revised Fig. 5a-c in the text to make it consistent with the terminology used in the XPS figure. Thank you again for your valuable suggestions. The revised figure is as follows:

Fig. 5 (a-c)

5. Given the mention of lattice oxygen participation during the oxidation process, why were the transition metals such as Co and Ni not oxidized to higher valence states, which could then contribute to the relaxed-stage leaching?

Response: Thank you for your insightful question. Regarding the valence state changes of Co and Ni, in our response to Reviewer 1's question 3, we acknowledged that the analysis of Co was incorrect and that the analyses of Ni and Mn were also incomplete. We apologize for these oversights and sincerely appreciate you bringing these critical issues to our attention. We have conducted a thorough reanalysis and revised the manuscript accordingly. The revised text is shown in line 287 as follows:

"The EELS analysis results are summarized in Fig. 5a-c and Supplementary Fig. 10.

Throughout the entire two-stage continuous oxidation lithium leaching process, the L_3 and L_2 peaks of Ni blue shifted, with peak intensity ratio decreased from 3.33 to 2.46, confirming continuous Ni oxidation (Fig. 5a). Co, however, exhibited a more complex behavior (Fig. 5b). The L_3 and L_2 peaks showed significant blue shifts in Stage I but then red shifts in Stage II. When viewed alongside the Co XPS data (Fig. 5e), this suggests that Co undergoes an initial reduction followed by subsequent oxidation. The initial reduction is linked to the oxidation of lattice oxygen by high-valence Co under the high voltage of Stage I.^{31,32} Notably, the L_3/L_2 ratio did not follow the same simple trend (decreasing from 2.45 to 2.21, then increasing to 2.33). This complex pattern is likely due to the simultaneous occurrence of sequence redox reaction and local structure evolution for Co, as has been reported in battery research community by Jiang³³ and Guo³⁴. Finally, Mn remained highly stable. Its L_3 and L_2 peak positions and L_3/L_2 ratio barely changed throughout the process (Fig. 5c). This stability can be attributed to the role of Mn in maintaining the structural integrity of the material, with minimal contribution to the charge storage.^{35"}

31 Shen, Y. et al. Sodium doping derived electromagnetic center of lithium layered oxide cathode materials with enhanced lithium storage. *Nano Energy* **94**, 106900 (2022).

32 House, R. A. et al. The role of O_2 in O-redox cathodes for Li-ion batteries. *Nature Energy* **6**, 781-789 (2021).

33 Jiang, Q. et al. Surface oxygenate species on TiC reinforce cobalt-catalyzed fischer-tropsch synthesis. *ACS Catal.* **11**, 8087-8096 (2021).

34 Guo, Y. et al. In situ exsolved CoFe alloy nanoparticles for stable anodic methane reforming in solid oxide electrolysis cells. *Joule* **8**, 2016-2032 (2024).

To more clearly illustrate the shifts in the L_3 and L_2 peak positions, we have revised the EELS spectra accordingly (Fig. 5a-c). The updated figure is presented below:

Fig. 5 Chemical and electronic state evolution of Ni, Co, and Mn during two-stage continuous oxidation lithium leaching process. **Ex-situ electron energy loss spectra of (a) Ni, (b) Co, and (c) Mn; Ex-situ X-ray photoelectron spectra of (d) Ni 2p, (e) Co 2p, and (f) Mn 2p; (g) XPS spectra of O1s; (h) O K-edge of EELS spectra; (i) Raman spectra.**

In summary, despite the complex nature of the redox process, it is evident that the electrochemical dual-oxidation (Stage I) involves the oxidation of lattice oxygen (O^{2-} to O^{n-} , where $n < 2$), alongside the concurrent oxidation of both Ni and Co. However, Co exhibits unique behavior: the electronic energy levels of Co^{3+} overlap with those of lattice oxygen, and under applied voltage, Co^{3+} facilitates the oxidation of lattice oxygen, resulting in the generation of Co^{2+} . This phenomenon has been frequently reported in studies of lithium-rich cathode materials, such as those by House et al. (2021) in Nature Energy (<https://doi.org/10.1038/s41560-021-00780-2>) and Qin et al. (2022) in Advanced Energy Materials (<https://doi.org/10.1002/aenm.202201549>). Specifically, in stage I, the oxidation of Co^{2+} and the reduction of Co^{3+} are competing reactions; because the reduction of Co^{3+} is stronger than the oxidation of Co^{2+} , the overall valence

state of Co decreases, as observed in both the Co EELS and XPS spectra at stage I.

6. In Supplementary Table 9, the units (\$) should be included in the last three rows (Total cost, Product revenue, and Total gross profit).

Response: Thank you for your helpful suggestion. We have added the missing unit (\$) to Supplementary Table 9.

7. Could the authors elaborate on how the two-stage leaching process would be integrated into a continuous mode for future industrial-scale applications?

Response: We sincerely thank you for raising this insightful question. We actually launched a startup company to commercialize our technique. In response to this comment, we have added a detailed discussion and experimental validation regarding the continuous and scaled-up application of the two-stage leaching process in the revised manuscript (Section 2.9 of the main text and Supplementary Note 8). A summary is provided below.

1. Design of the continuous system: We designed and fabricated a customized electrolysis system capable of processing a 500-gram batch of spent NCM111 powder (Fig. 7a-b, Supplementary Fig. 15a). The system employs multiple electrolytic cells in parallel and cost-effective titanium-based current collectors. This modular design enables a straightforward transition from batch to continuous operation through the replication of reaction modules, complemented by an automated material handling system.

2. Continuous process flow: As illustrated in Fig. 7a-b, the integrated continuous process comprises the following steps: 1) Stage I (EDO): Electrochemical dual-oxidation is conducted at a controlled voltage (2.5 V) for 70 minutes. 2) Stage II (Soaking-relaxation): The electrolytic cells containing the processed material are transferred from the EDO room to the soaking room for a 4-hour, energy-input-free (O^{n-} -driven) ion exchange reaction. 3) Product recovery: High-purity (>99.5%) Li_2CO_3 is precipitated and recovered from the resulting leachate, while the delithiated cathode material can be directly sold or subjected to further processing.

3. Scalability and Economic Assessment: Based on scaled-up experimental data, we have evaluated the economic viability of processing one ton of spent NCM111 material per day (Supplementary Table 9). Compared to a single-stage process requiring continuous EDO, the two-stage approach reduces energy consumption by 49.78%,

resulting in \$142.84 in savings in electricity costs per ton of material processed. The estimated gross profit reaches \$644.16 per ton of material, demonstrating strong economic potential for industrial-scale applications.

4. Specifics of integration into continuous mode: The process can be integrated into a continuous operation mode via a conveyor belt system, following the path ①→②→③→④→① as illustrated in Fig. 7b.

Fig. 7 (a) Process flow diagram for Li₂CO₃ recovery; (b) Schematic diagram of the continuous-flow system used for Li₂CO₃ recovery.

Supplementary Fig. 15 (a) Photographs of the customized equipment and electrolytic cell.

Finally, thank you for all the suggestions provided above. These comments have helped us improve the logic and clarity of the manuscript, as well as enhance the overall quality of the research. Your expert input and valuable time have been crucial for further

refinement of this paper.

Point-by-point Response Letter

Dear Respected Editor and Reviewers,

We sincerely thank you for reviewing our manuscript "**Maximizing energy utilization and lithium leaching efficiency via sequential electrochemical dual-oxidation and soaking-relaxation**" [NCOMMS-25-43469A] and for your valuable feedback. Your suggestions have greatly enhanced the quality of this work. Below is our response to your final comments.

Reviewer #1 (Remarks to the Author): The authors have solved the proposed issues; thus, I recommend the acceptance of this manuscript.

Response: Thank you for confirming that we have adequately addressed your previous concerns and for recommending acceptance of our manuscript. We are delighted to receive your positive assessment.

Reviewer #3: (Remarks to the Author): For my part, all my concerns have been addressed. I have also reviewed the questions raised by reviewer #2 as well as the authors' response. I feel that these concerns have also been satisfactorily addressed. Therefore, I suggest the publication of this manuscript in its current form.

Response: Thank you for your positive evaluation of our revisions. We especially appreciate that you also reviewed the issues raised by Reviewer #2 and our responses. We are glad that you found all concerns satisfactorily resolved and thank you for recommending publication in its current form.

Once again, we express our deep gratitude for your time, expertise, and constructive input.

Best wishes.